# Translation and Validation of the Volunteer Functions Inventory (VFI) among the General Dutch Older Population

**DOI:** 10.3390/ijerph16173106

**Published:** 2019-08-26

**Authors:** Jacobien Niebuur, Aart C. Liefbroer, Nardi Steverink, Nynke Smidt

**Affiliations:** 1Department of Epidemiology, University of Groningen, University Medical Center Groningen, PO Box 30.001, 9700 RB Groningen, The Netherlands; 2Netherlands Interdisciplinary Demographic Institute, PO Box 11650, 2502 AR The Hague, The Netherlands; 3Department of Sociology, Vrije Universiteit Amsterdam, De Boelelaan 1081, 1081 HV Amsterdam, The Netherlands; 4Department of Sociology, University of Groningen, Grote Rozenstraat 31, 9712 TG Groningen, The Netherlands; 5Department of Health Psychology, University of Groningen, University Medical Center Groningen, Antonius Deusinglaan 1, 9713 AV Groningen, The Netherlands; 6Department of Geriatrics, University of Groningen, University Medical Center Groningen, PO Box 30.001, 9700 RB Groningen, The Netherlands

**Keywords:** voluntary work, Volunteer Functions Inventory, motivations, cross-cultural, psychometric properties, validation, older adults

## Abstract

Improvement of volunteering rates in the Netherlands is important because increased productivity among older adults would contribute to societal sustainability in the light of population aging. Therefore, a better understanding of volunteer motivations of Dutch older adults is needed. The Volunteer Functions Inventory (VFI) for assessing volunteer motivations has good psychometric properties and is adapted to several languages, but no validated Dutch translation yet exists. The aim of the current study is to validate the VFI for use in the Dutch older population (60 years and over). The Dutch-translated VFI (6 scales, 30 items) is included in the Lifelines ‘Daily Activities and Leisure Activities add-on Study’, which was distributed among participants aged 60 to 80. Exploratory Factor Analysis (EFA) and Confirmatory Factor Analysis (CFA) are performed to assess the validity of the translated VFI. Internal consistency is assessed by computing Cronbach’s *α*’s. Results of the EFA (*N* = 4208) point towards a six-factor solution with a nearly perfectly clean structure. Deletion of three problematic items results in a clean factor structure. CFA results indicate moderate model fit (RMSEA = 0.06, CFI = 0.90, TLI = 0.89). Cronbach’s *α*’s (0.78 to 0.85) indicate good internal consistency. Goodness-of-fit indices are sufficient and comparable to those obtained in the validation of the original VFI. The current study provides support for use of the Dutch-translation of the VFI (6 scales, 27 items) to assess volunteer motivations among Dutch volunteers aged 60 years and over.

## 1. Introduction

In the Netherlands, many adults participate in voluntary work. Dutch individuals participate in a wide range of volunteer activities, among which voluntary activities for youth organizations, social and legal aid organizations, political parties, care and nursing organizations, and cultural associations are the most dominant [1]. The prevalence of participation in voluntary work among adults is estimated to be 33%, with the highest (38%) prevalence among adults aged 60 to 64 years [2]. The prevalence of volunteering among older adults in the Netherlands is comparable to volunteering rates in several Northern European countries (i.e., Sweden, Denmark), which are among the highest within Europe [3,4]. Participation in voluntary work carries various benefits, such as for the volunteering individual, recipients of voluntary work, the organizations volunteered for and societies as a whole [5]. Not only is participation in voluntary work positively associated with both physical [6,7] and mental health [7,8], social solidarity and individuals’ involvement in society [9], it also carries economic benefits [10]. Given that the proportion of older adults is increasing due to population aging, increasing volunteering, especially among older adults, could be beneficial for creating a sustainable society. In the Netherlands, the proportion of older adults (aged 65 years and over) is expected to rise from 19% in 2017 to 26% in 2040 [11].

Knowing the factors that influence participation in voluntary work is an important first step in studying the antecedents of participation in voluntary work. Results from a recent systematic review and meta-analysis showed that higher socioeconomic status, being married, larger social network size, church attendance, and previous volunteer experiences are positively related to volunteering whereas age, transitions into parenthood and functional limitations are inversely related [12]. However, in order to identify potential volunteers, it is also important to understand the underlying motives for people to participate in voluntary work. Why do some people who possess the resources to volunteer and lack the constraints decide to volunteer whereas others do not? And why do some volunteers continue their voluntary activities for a longer period of time while other volunteers quit? As the decision to start volunteering often involves actively seeking for volunteering opportunities, it raises the question why people would do so despite all the time and effort required [13].

Research on volunteer motivations is often based on functional theories originally formulated by Katz (1960) and Smith et al. (1956), which state that the same actions could serve different psychological functions for different individuals [13]. The strategy of functional analysis of volunteerism is adopted by Clary et al. (1998), proposing that “acts of volunteerism that appear to be quite similar on the surface may reflect markedly different underlying motivational processes” [13] (p. 1517). The application of the functional analysis to the field of volunteer motivations explains the underlying reasons for engaging in voluntary work, by clarifying the functions that can be served by participation in voluntary work [13]. In the Volunteer Functions Inventory (VFI), which is a measurement instrument aimed at assessing the motives to volunteer [13], six motivational functions are proposed. These motivational functions [13] (p.1518) are (a) the Values function: The opportunities that volunteerism provides for individuals to express values that are important to the self, related to altruistic and humanitarian concerns for others, (b) the Understanding function: The opportunity for volunteers to gain and sustain knowledge, skills and abilities, (c) the Social function: Volunteering offers opportunities to improve social relationships, it helps individuals to fit in and get along with social groups that are important for them, (d) the Career function: Participation in voluntary work may increase future job opportunities, (e) the Protective function: Participation in voluntary work can protect oneself from negative feelings about oneself, it can help the individual to eliminate negative aspects surrounding the ego and (f) the Enhancement function: By participation in voluntary work the individual can enhance his/her self-esteem, it centers on ego growth and development.

The Volunteer Functions Inventory (VFI) is the most widely used instrument for assessing volunteer motivations [14]. The VFI is widely used because of its theoretically sound basis as well as its good psychometric properties [14]. Originally, the VFI was validated in a sample of middle-aged Americans who were actively involved as volunteers, and cross-validated in a sample of American university students with and without volunteer experience [13]. The applicability of the VFI as a measurement tool for assessing volunteer motivations has been confirmed in multiple volunteer settings and different age groups [15]. Moreover, the VFI has been translated and validated in the German [16] and Chinese [17] languages. Several VFI items have been translated into Dutch [18]. However, the total VFI has not been translated, nor validated in the Dutch language yet. The current study, therefore, aims to translate the VFI to the Dutch language and to evaluate the psychometric properties of the Dutch-translated VFI within a sample of current volunteers aged 60 to 80 years in order to assess its cross-cultural validity.

## 2. Materials and Methods

### 2.1. Study Design and Participants

The Lifelines cohort study is a multi-disciplinary prospective population-based cohort study examining in a unique three-generation design the health and health-related behaviors of 167,729 persons living in the North of the Netherlands. It employs a broad range of investigative procedures in assessing the biomedical, socio-demographic, behavioral, physical, and psychological factors which contribute to the health and disease of the general population, with a special focus on multi-morbidity and complex genetics. The study profile of Lifelines is described elsewhere [19,20]. Briefly, participants were recruited between 2006 and 2013. Inhabitants (aged 25 to 50 years) of the three Northern provinces of the Netherlands were invited by their general practitioners (GPs) if they met eligibility criteria. Subsequently, respondents’ family members were invited, leading to a unique three-generation design. Additionally, inhabitants of the Northern provinces of the Netherlands could self-register via the Lifelines website. Baseline assessment (T1), consisting of physical examination, collecting fasting blood and urine samples, interviews, and self-report questionnaires, was conducted between 2006 and 2013. Participants were followed-up every 1.5 years by additional questionnaires and every five years by a physical examination. The Lifelines Cohort Study is approved by the medical ethical committee of the University Medical Center Groningen, the Netherlands. All participants signed an informed consent form. Lifelines is a facility that is open for all researchers. Information on the application and data access procedure is summarized on www.lifelines.nl.

#### Lifelines Daily Activities and Leisure Activities Add-On Study (Lifelines DALAS)

All adults aged 60 to 80 who completed one of the two parts of the fourth Lifelines questionnaire at least six months before the start of the Lifelines ‘daily activities and leisure activities add-on study (Lifelines DALAS)’ were invited to participate in Lifelines DALAS. The Lifelines DALAS questionnaire consists of a broad range of measures related to health, quality of life and lifestyle as well as a broad range of questions assessing the daily activities (i.e., employment, providing informal care and voluntary work, taking care of grandchildren) and leisure activities (i.e., sports, cultural activities, traveling, social contacts) of participants. A full section of the questionnaire is devoted to participation in voluntary work, containing questions about current and former participation in voluntary work, the frequency, duration, intensity and type of volunteering as well as the motives underlying volunteering. Respondents who indicated to currently participate in voluntary work were asked to fill-out the translated version of the VFI.

### 2.2. Translation of the VFI

In the Volunteer Functions Inventory (VFI), each of the six functions is represented by one factor in the measurement instrument. The VFI consists of 30 separate items and each factor is represented by a total of five items. The items are introduced by the following phrase ‘Please indicate how important or accurate each of the 30 possible reasons for volunteering was for you in doing volunteer work’. Each item is rated on a seven-point Likert scale where item-score 1 represents ‘not at all important/accurate’ and item-score 7 represents ‘extremely important/accurate’.

The forward-backward-translation method [21] is used to translate the original English VFI into Dutch. Briefly, the forward-backward-translation method consists of the following steps: (a) forward translation by two independent translators, (b) review of translation by experts, (c) back translation by two independent translators who are native in English, (d) review of translation by experts, (e) production of final translation. The two independent translators who perform the forward translation are both native in Dutch and fluent in English. One of them is a professional translator without knowledge of the instrument and the other translator has content experience.

### 2.3. Statistical Analysis

Cross-cultural validation of the final translation of the VFI was performed within the sample of current volunteers who filled out the VFI in Lifelines DALAS. In order to check the validity of the cross-culturally adapted VFI, we needed to assess its construct validity [22]. Factor analysis was used to assess measurement invariance. The idea of performing factor analysis for cross-cultural validation purposes was to assess whether the translated items had the same meaning as the original items. If a translated item did not load on the intended factor, the meaning of the translated item was different from the meaning of the original item, either as a result of improper translation or cultural differences [22]. Reliability of the scales was assessed by Cronbach’s *α*. First, Exploratory Factor Analysis (EFA) was conducted and the reliability of the resulting scales was assessed. Subsequently, the resulting scale was assessed by means of Confirmatory Factor Analyses (CFA) in STATA.

#### 2.3.1. Exploratory Factor Analysis

Exploratory Factor Analysis (EFA) was performed in SPSS (IBM, Armonk, NY, USA). We used either Maximum Likelihood (ML) estimation or Principal Axis Factoring (PAF) depending on whether the data were generally normally distributed (ML) or significantly non-normally distributed (PAF), respectively. These methods reveal the factor structure in the data caused by latent variables underlying the different factors [23]. Direct oblique rotation was used (using the default delta (0) and kappa (4) values in SPSS) if the correlation between all factors was >0.32, otherwise Varimax rotation was used [23].

Items that did not correlate 0.20 or higher with any of the other items were to be deleted immediately [22]. Items with correlations above 0.90 were carefully considered because of potential multicollinearity problems. In deciding on the number of factors to retain for rotation, we followed the guidelines proposed by Costello and Osborne [23]. We started by setting the number of factors to retain equal to the number of factors in the original scale. Therefore, we started with the preselected six-factor solution. If the scree-test suggested a different number of factors compared to the preselected factor solution, we subsequently reran the analysis for (a) the number of factors suggested by the scree-test, (b) the numbers below and above the projected number based on the a-priori factor structure and (c) the numbers below and above the number of factors suggested by the scree-test. Only factor loadings >0.3 were retained. Items with factor loadings <0.50 should be considered carefully too. After rotation, we compared the tables with item loadings to select the “cleanest” factor structure. The cleanest factor structure was defined as the factor structure with item loadings above 0.30, no or few items with cross-loadings (cross-loadings of >0.30 are not desirable) and no factors with fewer than three items.

#### 2.3.2. Reliability

The degree of interrelatedness among the items within factors was analyzed by assessing Cronbach’s *α*, in order to check whether or not the items within one factor measured the same construct. The underlying principle of examining the internal consistency of scales by means of Cronbach’s *α* was to split the items in half and check if the mean value of the correlations between the scores of each two half-scales was sufficient [22]. Cronbach’s *α*’s should preferably have a value above 0.70 [24]. Item-total correlation (which gives an indication of whether the items discriminate respondents on the construct under study) should be larger than 0.30 [22]. If it was lower, deletion of the item was considered.

#### 2.3.3. Confirmatory Factor Analysis

Confirmatory Factor Analysis (CFA) was conducted in STATA by means of Structural Equation Modeling (SEM). Model fit was assessed by means of several goodness-of-fit indices. We used the Root Mean Square Error of Approximation (RMSEA) and the Standardized Root Mean Squared Residual (SRMR) to assess the absolute fit (i.e., how strongly a hypothesized model deviates from a perfect model) and the Comparative Fit Index (CFI), Tucker–Lewis Index (TLI) and the Coefficient of Determination (CD) to assess incremental fit (i.e., the fit of a hypothesized model compared to the fit of a baseline model). For Maximum Likelihood (ML) estimation, Hu and Bentler suggest that an RMSEA < 0.06, SRMR < 0.80 and a CFI, TLI and CD > 0.95 indicate a relatively good model-data fit [25].

### 2.4. Sensitivity Analysis

STATA uses Maximum Likelihood (ML) estimation in SEM analyses by default. This estimation is based on the normality assumption. Because the data from the VFI are ordered categorical data (Likert scales are used) and not continuous, applying ML could result in biased parameter estimates, inaccurate standard errors and a misleading *χ*^2^statistic [26]. An option to overcome estimation problems due to non-normality, is to use the ADF (Asymptotically Distribution-Free) estimation method which is based on the weighted least squares estimator (WLS). To assess whether the CFA results were robust for the estimation method, SEM based on ADF estimation was conducted as a sensitivity analysis.

## 3. Results

### 3.1. Translation of the VFI

Forward translation was performed by the authors of the current study (i.e., the informed translation) and, independently, by a translator working at the University Translation Service of the University of Groningen (i.e., the uninformed translation). Next, our research team discussed discrepancies between the two forward translations and combined the two into a synthesized Dutch version. Subsequently, back translation of this version was done independently by two native English speakers, both working at the University Translation Service. A couple of back-translated items were slightly different. Our research team combined these two versions into a single version selecting the items of each translation that, in our opinion, most accurately captured the meaning of the original items. We sent this final version to the authors of the original VFI who suggested reconsidering two items (items 6 and 16). We concluded that the differences between the original items and the translated items were subtle and mainly reflected differences between linguistic and national contexts. Therefore, no further adaptation of the translated items was deemed necessary.

The answer scale in the translated version differs from the original one. A pilot study (*N* = 8, age range 61–74 years) was conducted in which the original answer scale was tested. Respondents indicated difficulties with the original answer scale (‘important or accurate for me’), because some items were more important than accurate for them or the other way around. For that reason, the answer scale was adapted. Each item was again rated on a seven-point Likert scale where item-score 1 represents ‘does not apply at all’ and item-score 7 represents ‘applies in full’. The final Dutch translation of the VFI which was used in the validation process is provided in Appendix A.

### 3.2. Characteristics of the Study Population and the Measurement Scale 

A total of 15,655 participants were invited to participate in the Lifelines DALAS study. A total of 7639 participants filled out the questionnaire (response rate of 49.0%), with volunteer status being provided by 7612 respondents (99.6%). Of these, 4208 respondents (55.3%) indicated to participate in voluntary work at the time of completing the questionnaire. This volunteer subsample was selected for the current study and all respondents in this subsample were asked to fill out the Volunteer Functions Inventory (VFI). The mean age of the respondents in the volunteer sample was 67.06 years (*SD* = 4.73), about half of the sample consisted of females (*N* = 2123, 50.5%), and the vast majority was married or cohabiting (*N* = 3538, 86.5%). Moreover, the education level of the respondents was high, as 65.7% of the respondents had achieved upper secondary or tertiary education. Finally, the majority of the sample was retired (*N* = 2927, 69.7%) (see Table 1). Scale descriptive statistics for the VFI are provided in Table 2. As Table 2 shows, the percentage of missing values on the items of the VFI scale was low, ranging from 1.2% (items 1, 5 and 26) to 1.8% (items 23 and 30). The majority of the VFI items responses were highly non-normally distributed. Mean item response scores ranged from 1.39 (*SD* = 1.10) (item 1, Career) to 5.38 (*SD* = 1.59) (item 8, Values). The mean (*M*) scores on the six factors ranged from 1.49 on Career, to 4.78 on Values (from low to high Career (*M* = 1.49, *SD* = 0.93), Protective (*M* = 2.01, *SD* = 1.14), Social (*M* = 2.56, *SD* = 1.25), Enhancement (*M* = 3.34, *SD* = 1.47), Understanding (*M* = 3.44, *SD* = 1.53), Values (*M* = 4.78, *SD* = 1.31)).

### 3.3. Exploratory Factor Analyses

The distribution of the data of all VFI items was non-normal, therefore, Principal Axis Factoring (PAF) with oblique rotation was used to assess the number of factors. In the preselected six-factor solution, the first six components had eigenvalues above 1, indicating the presence of six factors underlying the responses to the VFI. For the first six factors resulting from the current analysis, the eigenvalues and cumulative percentage variance (presented between brackets) were 9.67 (32.2%), 2.87 (41.8%), 1.70 (47.5%), 1.50 (52.5%), 1.39 (57.1%), and 1.06 (60.6%). The vast majority of the items from the individual scales loaded on their intended factor without cross-loadings. Three items did not load on their intended scales. Item 11 (‘Doing volunteer work relieves me of some of the guilt over being more fortunate than others’) loaded on the Career factor instead of on the Protective factor. Item 22 (‘I can do something for a cause that is important to me’) did not have a factor loading of >0.3 on any of the factors. Interestingly, item 29 (‘Volunteering is a way to make new friends’) loaded on the Understanding factor instead of on the intended factor Enhancement. This was also the case in both the validation study (study 1) as well as the cross-validation (study 2) of the original VFI [13]. Because item 29 persistently loaded on another factor (Understanding) than on the intended factor (Enhancement), based on theoretical reasoning in both the first and second original validation studies [13] as well as in our validation sample, item 29 was deleted.

Although the eigenvalues pointed towards a six-factor solution, the scree plot seemed to point towards either a two-factor or a five-factor solution. Therefore, Principal Axis analyses were performed for a two-factor, three-factor, four-factor, five-factor and seven-factor solution too. None of them resulted in a nearly clean structure, confirming the idea that there are six factors underlying the responses. The six-factor analysis resulted in a nearly perfectly clean structure: all resulting factors had more than three items, there were no cross-loadings and the majority of the items loaded on their intended factors. Item 22 was deleted because it loaded too low (<0.3) on any of the factors. Item 11 had a factor loading of <0.4 (0.33), whereas loadings >0.5 were preferred. Therefore, deleting this item was considered next, assessing the internal consistency of the scales. The pattern matrix containing all scales, items and factor loadings (>0.30) is presented in Table 3.

### 3.4. Internal Consistency

The internal consistency of each of the VFI scales was assessed by computing Cronbach’s α coefficients. Coefficients were relatively high but slightly below those of the original VFI scale. The resulting Cronbach’s α’s: Protective 0.81, Values 0.78, Career 0.84, Social 0.78, Understanding 0.85 and Enhancement 0.85. For each of the scales, Cronbach’s α was also computed after eliminating items with factor loadings <0.50. Eliminating item 11 slightly improved the Cronbach’s α coefficient of the Career factor (from 0.84 to 0.85). Given that the factor loading for item 11 on the Career factor was only 0.33 and Cronbach’s α improved by eliminating this item, we decided to delete item 11. For all items, item-total correlations were far above 0.30, ranging from 0.46 to 0.75.

### 3.5. Confirmatory Factor Analysis

Deletion of a couple of problematic items in the EFA (items 11, 22 and 29) resulted in a clean factor structure, confirming the presence of six distinct factors for the Dutch-translated version of the VFI. The resulting scale from the EFA, that was used for validation by means of CFA, consisted of the six subscales proposed by the original VFI and a total of 27 items: Understanding (items 12, 14, 18, 25, 30), Career (items 1, 10, 15, 21, 28), Values (Items 3, 8, 16, 19), Protective (items 7, 9, 20, 24), Social (items 2, 4, 6, 17, 23) and Enhancement (items 5, 13, 26, 27). Fit indices from the confirmatory factor analysis resulted in a moderate goodness of fit (RMSEA = 0.06, CFI = 0.90, TLI = 0.89), see Table 4. The *p*-value for the *χ*^2^ statistic (*χ*^2^(309) = 5366.59) was significant (*p* < 0.001) indicating model-data misfit, but this may have been due to the large sample size potentially biasing the *χ*^2^ estimate. Goodness-of-fit indices were sufficient and comparable to the goodness-of-fit indices obtained in the validation of the original VFI scale [13]. Scale correlations are presented in Table 5. Our findings offer support for the Dutch translation of the VFI as a measure of volunteer motivations among samples of volunteers aged 60 years and over.

### 3.6. Sensitivity Analysis

Using Maximum Likelihood (ML) is especially problematic when “the number of categories is below five and the categorical distribution is highly asymmetric” [22] (p.2). As Table 2 shows, although all VFI items have seven categories, and the sample size was relatively large (*N* = 4208), the distribution of the items was in general highly asymmetric and using ML could have posed problems. Therefore, SEM based on ADF estimation was conducted as a sensitivity analysis in order to assess whether the CFA results were robust to the estimation method. The fit indices of the sensitivity analysis can be found in Table 4. As Table 4 shows, the RMSEA index slightly improved, but the CFI and TLI values slightly worsened. Factor loadings patterns resulting from ML estimation and those from ADF estimation were roughly comparable (see Appendix A).

## 4. Discussion

The aim of the current study was to assess the cross-cultural validity of the Dutch-translated VFI. The findings provide support for use of the translated VFI (6 scales, 27 items) to assess the motivations to volunteer among Dutch samples of volunteers aged 60 years and over.

EFA results showed that 27 out of the 30 items loaded on their intended scale and had no cross-loadings on other factors. Three items (items 11, 22 and 29) of the original scale did not perform well in the Dutch version. The cross-cultural validation, therefore, resulted in the deletion of three problematic items, leaving us with a total of 27 items. Items 11 and 29 are deleted because they loaded on another than the intended factor. Item 11 (‘Doing volunteer work relieves me of some of the guilt over being more fortunate than others’) is part of the Protective factor but loaded (weakly) on the Career factor. Although in the original validation studies [13] this item did load on the intended factor, its loading was relatively low (0.43) [13]. Item 29 (‘Volunteering is a way to make new friends’) is part of the Enhancement factor. However, not only in the current validation study but also in the original validation studies 1 and 2 [13], item 29 persistently loaded on the Understanding factor instead of on the Enhancement factor, as theoretically intended. From an Enhancement point of view, this item seems to focus on the outcome of volunteering: by volunteering one can make new friends, which in turn can increase positive affect and ego growth. From an Understanding point of view, however, this item seems to focus on the means to make new friends: by volunteering social skills can be improved, which in turn can help in making new friends. The focus on the social skills obtained from volunteering could be an explanation for the persistent loading of item 11 on the Understanding factor. Finally, item 22 (‘I can do something for a cause that is important to me’) is part of the Values factor but is deleted because we found a very low (<0.3) factor loading, which could be related to the translation of the item. In the original item, the focus is on the importance of the cause, whereas in our Dutch translation the focus could be perceived to be on the importance of the cause *for* the respondent, rather than the importance of the cause as perceived by the respondent.

In the current study, factor mean scores ranged from 1.49 (*SD* = 0.93) for the Career factor, to 4.78 (*SD* = 1.31) for the Values factor. The sequence of the importance of volunteer functions is comparable to those reported for the samples of the original validation study (study 1) and cross-validation (study 2) of the VFI [13], with lower scores on the Career, Protective and Social factors and higher scores on the Enhancement, Understanding and Values factors. However, factor mean scores are substantially lower in the current study than in the original validation studies. A first explanation could be that this is due to our adaptation of the answer score. ‘Applies to me’ in the translated version differs from ‘important or accurate for me’. However, ‘applies to me’ is less strong of an expression than ‘important or accurate for me’. Therefore we would expect the item mean scores to be a bit higher than in the original validation studies [13], rather than lower. A second, more substantive explanation could be that culturally related differences in personality influence the rating of the items. Dutch people are less extravert than Americans [27], and could, therefore, have a tendency to express themselves less strongly than Americans do. However, acquiescence bias could also play a role. Cultural differences exist in the tendency to agree with test items [27]. It could be that Americans, in general, have a stronger tendency to agree with test items than Dutch people do. The study of Schmitt et al. [27] shows that American respondents do not only score higher on extraversion than Dutch respondents, but also on the other four Big Five personality traits. The higher scores on volunteer motives could either be a result of the higher extraversion of American as compared to Dutch respondents, or both the higher scores on volunteer motives in the current study and the higher extraversion score in the study of Schmitt et al. [27] could result from a stronger tendency to agree with test items by Americans as compared to Dutch.

The Career factor seems to be much less important in the current sample than in the samples used in the original validation studies. In our study, the Career factor has the lowest mean score (1.49), which can be explained by the fact that the majority of the sample (69.7%) was retired and the remainder of the sample was relatively close to retirement, as our sample includes adults aged 60 years and over. Therefore, our respondents are probably less focused on their careers than the younger people in the samples used for the original validation studies, containing middle-aged adults (study 1) and high school students (study 2). Similar factor mean scores were obtained in the study by Okun, Barr, and Harzog [28] who validated the VFI in two samples of American older adults (1.36 in sample 1 (96% ≥ 60 years of age, 1.48 in sample 2 (88% ≥ 60 years of age). Age is demonstrated to be inversely related to the Career motive [28].

The model fit in the current study is moderate (RMSEA = 0.06, CFI = 0.90, TLI = 0.89). Overall, though, results from the EFA and CFA analyses suggest good validity of the Dutch VFI. In our opinion, model fit indices should not be used as sole indication for the validity of a measurement scale but should rather be evaluated in combination with other indications of validity, and cut-off values for model fit should be used as a guideline. EFA results clearly pointed towards a six-factor solution containing the six factors distinguished in the original scale, with 27 out of 30 items loading on the intended factors with quite high factor loadings. Our results are in line with those of studies in other countries in which the VFI scale was validated. In the study samples used by Clary et al. [13] and Okun, Barr and Harzog [28] the presence of six distinct factors is consistently demonstrated and the pattern matrices are comparable. The CFA fit measures in the original validation studies demonstrated comparable model fit (validation study 1: RMSres = 0.057, GFI = 0.91, NFI = 0.90, validation study 2: RMSres = 0.065, GFI = 0.89, NFI = 0.88) [13]. Moreover, CFA analyses by Okun, Burr and Herzog [28] resulted in similar fit indices as in the first sample (RMSEA = 0.06, CFI = 0.90, IFI = 0.90).

Sensitivity analyses were conducted by performing SEM analyses based on ADF, because the distribution of the items is in general highly asymmetric and as a result, using ML can give problems. Results of our sensitivity analysis show that the RMSEA measure slightly improves, but CFI and TLI values slightly worsen by changing the method from ML to ADF. It is unclear whether better results for the RMSEA measure are an indication of better model specification or just the result of changing the estimation method [26]. Comparing factor loadings from ML estimation with factor loadings from ADF estimation shows that the patterns of factor loadings are roughly comparable. All factor loadings from ADF are of the same sign as those from ML, and most factor loadings are of comparable magnitude. Although the results from ADF estimation are inconclusive, the comparability of the factor loadings and pattern matrices between the two estimation methods provides some reassurance for the robustness of the results for the estimation method.

### 4.1. Study Strengths and Limitations

To our knowledge, this is the first validation of the VFI in the Dutch population. Among the strengths of this study are the large sample size (*N* = 4208), the use of a random sample of the Dutch older population, and the low amount of missing values (1.2% to 1.8% per item). The current study has some limitations too. The study sample is not fully representative of the Dutch population aged 60 and over, as the Lifelines population, in general, is relatively highly educated compared to the general population [20]. Besides, the change in the answer scale could have had an influence on the rating of the items in the Dutch-translated VFI. Item mean scores may probably be a bit higher than they would have been if we had adopted an exact translation of the original answer scale. Finally, no test-retest reliability has been performed.

### 4.2. Implications for Future Research

The current study demonstrates that the Dutch version of the VFI (6 scales, 27 items) is a valid instrument for assessing volunteer motives among Dutch volunteers aged 60 years and over. Volunteer motives of Dutch older volunteers can now be assessed by making use of a validated measurement instrument, enabling researchers to assess and compare the motives of Dutch older volunteers. The possibility to improve the knowledge and understanding of volunteer motives in Dutch older volunteers could be helpful for volunteer retention practices. Voluntary work could be arranged in such a way that volunteer activities more closely address the motives driving Dutch older volunteers, potentially improving retention rates.

The items of the Career subscale are skewed in the current study population. Therefore, in using the Dutch VFI in multivariate analyses, the skewness should be taken into account, for example by using a transformation that makes the scale less skewed. In future research, performing a test-retest reliability analysis of the Dutch VFI in a distinct sample of Dutch adults aged 60 to 80 would corroborate the reliability of the scale. Moreover, as the translated VFI is validated within a sample of currently volunteering individuals, it would be useful to test the validity of the Dutch VFI in Dutch samples of current non-volunteers too, because then volunteer motives can be assessed in potential volunteers. Furthermore, inclusion of the validated Dutch version of the VFI in the Lifelines Cohort Study would provide opportunities to assess volunteer motivations in relation to factors such as socio-demographics, and actual participation in voluntary work (e.g., frequency, hours volunteered, duration), and ultimately, associations with outcomes related to societal sustainability. Finally, it would be interesting to examine whether different population groups (i.e., based on differences in culture, religious affiliation, immigrant status) expect the same gains from volunteering.

## 5. Conclusions

In summary, we have demonstrated that the translated and adapted Dutch version of the VFI, consisting of 6 scales and 27 items, is a valid instrument for assessing volunteer motives among Dutch adults aged between 60 and 80 years of age who are currently participating in voluntary work.

## Figures and Tables

**Table 1 ijerph-16-03106-t001:** Sample background characteristics (*N* = 4208).

Background Characteristics	*N* (%) ^1^
Age (*M* (*SD*), range)	67.06 (4.73), 60–80
Gender (Female)	2123 (50.5%)
Educational attainment	
- Elementary	81 (2.0%)
- Lower secondary	1319 (32.3%)
- Upper secondary	1173 (28.7%)
- Tertiary	1512 (37.0%)
Marital status	
- Married/cohabiting	3538 (86.5%)
- Relationship not cohabiting	109 (2.6%)
- Single/no partner	459 (10.9%)
Employment status ^2^	
- Employed	1048 (24.9%)
- Retired	2927 (69.7%)
- Unemployed	158 (3.8%)
- Disabled from work	121 (2.9%)

^1^ All numbers are *N* (%), unless indicated otherwise. Percentages are valid percentages (excluding missing cases). ^2^ For the employment status variables, dichotomous measures are used (employed versus not-employed, retired versus not retired, unemployed versus not unemployed and disabled from work versus not disabled from work). The percentages in the table are based on these dichotomous variables and therefore do not add up to 100%. Some respondents do not belong to any of these four categories and others belong to several categories (for example, a respondent can both be employed and disabled from work for a certain percentage of his or her working hours).

**Table 2 ijerph-16-03106-t002:** Descriptive statistics for the Volunteer Functions Inventory (VFI) items.

Subscales	Items	Mean (*SD)*	Median	Skewness	Kurtosis	Missing Data (*N* (%))
Understanding	12. I can learn more about the cause for which I am working	2.84 (2.000)	2	0.601 (0.038)	−1.102 (0.076)	54 (1.3%)
	14. Volunteering allows me to gain a new perspective on things	3.76 (1.973)	4	−0.138 (0.038)	−1.297 (0.076)	60 (1.4%)
	18. Volunteering lets me learn things through direct, hands on experience	3.44 (1.941)	4	0.119 (0.038)	−1.307 (0.076)	69 (1.6%)
	25. I can learn how to deal with a variety of people	3.91 (2.007)	4	−0.198 (0.038)	−1.278 (0.076)	69 (1.6%)
	30. I can explore my own strengths	3.23 (1.967)	3	0.245 (0.038)	−1.313 (0.076)	74 (1.8%)
Career	1. Volunteering can help me to get my foot in the door at a place where I would like to work	1.39 (1.104)	1	3.175 (0.038)	9.978 (0.076)	52 (1.2%)
	10. I can make new contacts that might help my business or career	1.60 (1.312)	1	2.363 (0.038)	4.747 (0.076)	57 (1.4%)
	15. Volunteering allows me to explore different career options	1.45 (1.086)	1	2.802 (0.038)	7.673 (0.076)	59 (1.4%)
	21. Volunteering will help me to succeed in my chosen profession	1.45 (1.081)	1	2.788 (0.038)	7.652 (0.076)	69 (1.6%)
	28. Volunteering experience will look good on my résumé	1.57 (1.292)	1	2.495 (0.038)	5.513 (0.076)	73 (1.7%)
Values	3. I am concerned about those less fortunate than myself	3.88 (2.167)	4	−0.118 (0.038)	−1.436 (0.076)	56 (1.3%)
	8. I am genuinely concerned about the particular group I am serving	5.38 (1.587)	6	−1.254 (0.038)	1.055 (0.076)	52 (1.2%)
	16. I feel compassion toward people in need	4.87 (1.729)	5	−0.844 (0.038)	−0.094 (0.076)	58 (1.4%)
	19. I feel it is important to help others	5.34 (1.485)	6	-1.090 (0.038)	0.887 (0.076)	60 (1.4%)
	22. I can do something for a cause that is important to me	4.43 (2.034)	5	−0.535 (0.038)	−1.030 (0.076)	64 (1.5%)
Protective	7. No matter how bad I have been feeling, volunteering helps me to forget about it	2.60 (1.857)	2	0.785 (0.038)	−0.729 (0.076)	54 (1.3%)
	9. By volunteering I feel less lonely	2.22 (1.661)	1	1.215 (0.038)	0.315 (0.076)	56 (1.3%)
	11. Doing volunteer work relieves me of some of the guilt over being more fortunate than others	1.67 (1.282)	1	2.069 (0.038)	3.545 (0.076)	56 (1.3%)
	20. Volunteering helps me work through my own personal problems	1.85 (1.392)	1	1.757 (0.038)	2.335 (0.076)	69 (1.6%)
	24. Volunteering is a good escape from my own troubles	1.72 (1.319)	1	2.001 (0.038)	3.331 (0.076)	69 (1.6%)
Social	2. My friends volunteer	2.15 (1.667)	1	1.293 (0.038)	0.483 (0.076)	55 (1.3%)
	4. People I am close to want me to volunteer	1.69 (1.325)	1	2.090 (0.038)	3.652 (0.076)	54 (1.3%)
	6. People I know share an interest in community service	2.65 (1.794)	2	0.685 (0.038)	−0.829 (0.076)	56 (1.3%)
	17. Others with whom I am close place a high value on community service	3.57 (1.910)	4	0.013 (0.038)	−1.261 (0.076)	68 (1.6%)
	23. Volunteering is an important activity to the people I know best	2.72 (1.839)	2	0.679 (0.038)	−0.826 (0.076)	75 (1.8%)
Enhancement	5. Volunteering makes me feel important	2.84 (1.787)	2	0.512 (0.038)	−0.996 (0.067)	50 (1.2%)
	13.Volunteering increases my self-esteem	3.30 (1.928)	3	0.197 (0.038)	−1.279 (0.076)	53 (1.3%)
	26. Volunteering makes me feel needed	3.81 (1.894)	4	−0.176 (0.038)	−1.204 (0.076)	68 (1.6%)
	27. Volunteering makes me feel better about myself	3.39 (1.895)	4	0.108 (0.038)	−1.276 (0.076)	73 (1.7%)
	29. Volunteering is a way to make new friends	3.35 (1.951)	3	0.167 (0.038)	−1.294 (0.076)	64 (1.5%)

**Table 3 ijerph-16-03106-t003:** Exploratory Factor Analysis (EFA) results: Pattern matrix for the Dutch VFI (translated version of the original VFI) (Principal-Axis Factor Analysis, Oblique Rotation, Six factors pre-specified).

VFI-V Scale and Items	Factor					
	1	2	3	4	5	6
**1. Understanding**						
12. I can learn more about the cause for which I am working	0.446	-	-	-	-	-
14. Volunteering allows me to gain a new perspective on things	0.558	-	-	-	-	-
18. Volunteering lets me learn things through direct, hands on experience	0.661	-	-	-	-	-
25. I can learn how to deal with a variety of people	0.519	-	-	-	-	-
30. I can explore my own strengths	0.721	-	-	-	-	-
**2. Career**						
1. Volunteering can help me to get my foot in the door at a place where I would like to work	-	−0.635	-	-	-	-
10. I can make new contacts that might help my business or career	-	−0.703	-	-	-	-
15. Volunteering allows me to explore different career options	-	−0.822	-	-	-	-
21. Volunteering will help me to succeed in my chosen profession	-	−0.646	-	-	-	-
28. Volunteering experience will look good on my résumé	-	−0.618	-	-	-	-
**3. Values**						
3. I am concerned about those less fortunate than myself	-	-	0.575	-	-	-
8. I am genuinely concerned about the particular group I am serving	-	-	0.551	-	-	-
16. I feel compassion toward people in need	-	-	0.867	-	-	-
19. I feel it is important to help others	-	-	0.787	-	-	-
22. I can do something for a cause that is important to me	-	-	-	-	-	-
**4. Protective**						
7. No matter how bad I have been feeling, volunteering helps me to forget about it	-	-	-	0.370		-
9. By volunteering I feel less lonely	-	-	-	0.451		-
11. Doing volunteer work relieves me of some of the guilt over being more fortunate than others	-	−0.331	-	-		-
20. Volunteering helps me work through my own personal problems	-	-	-	0.823		-
24. Volunteering is a good escape from my own troubles	-	-	-	0.837		-
**5. Social**						
2. My friends volunteer	-	-	-	-	−0.592	-
4. People I am close to want me to volunteer	-	-	-	-	−0.483	-
6. People I know share an interest in community service	-	-	-	-	−0.730	-
17. Others with whom I am close place a high value on community service	-	-	-	-	−0.595	-
23. Volunteering is an important activity to the people I know best	-	-	-	-	−0.499	-
**6. Enhancement**						
5. Volunteering makes me feel important	-	-	-	-	-	0.610
13.Volunteering increases my self-esteem	-	-	-	-	-	0.676
26. Volunteering makes me feel needed	-	-	-	-	-	0.475
27. Volunteering makes me feel better about myself	-	-	-	-	-	0.574
29. Volunteering is a way to make new friends	0.612	-	-	-	-	-

**Table 4 ijerph-16-03106-t004:** Confirmatory Factor Analysis (CFA) results: Goodness of fit statistics.

Structural Equation Modeling (SEM) Based on Maximum Likelihood (ML) Estimation for the 6-Factor Solution with 27 items								
N	χ^2^	df	P	RMSEA	CFI	TLI	SRMR	CD
4010	5366.59	309	0.000	0.064	0.899	0.886	0.053	1.000
**Sensitivity Analysis: Structural Equation Modeling (SEM) based on ADF Estimation**								
4010	2771.59	309	0.000	0.045	0.740	0.705	0.092	1.000

**Table 5 ijerph-16-03106-t005:** Factor correlation matrix (6 factors, 27 items).

Subscales	1. Understanding	2. Career	3. Values	4. Protective	5. Social	6. Enhancement
1. Understanding	1.000	−0.326	0.438	0.387	−0.383	0.520
2. Career	−0.326	1.000	−0.072	−0.414	0.430	−0.289
3. Values	0.438	−0.072	1.000	0.194	−0.314	0.312
4. Protective	0.387	−0.414	0.194	1.000	−0.369	0.402
5. Social	−0.383	0.430	−0.314	−0.369	1.000	−0.406
6. Enhancement	0.520	−0.289	0.312	0.402	−0.406	1.000

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
