# Peer review of "Translation and Validation of the Volunteer Functions Inventory (VFI) among the General Dutch Older Population"

_ijerph, 2019, doi:10.3390/ijerph16173106_

Round 1
Reviewer 1 Report
Thank you for your reply to my comments. I do not have any additional comments on your manuscript.
Author Response
Thank you very much for review this paper again
Reviewer 2 Report
In this study, the authors translate the Volunteer Functions Inventory into Dutch and validate the translated scale using exploratory and confirmatory factor analysis with a population of volunteers aged 60-80. This article has the potential to make an important contribution as the first Dutch translation of the VFI, and a validation of the instrument in a new context.
I the introduction to the article, the authors do a nice job of framing the need for the study by focusing on the practical importance of learning more about what motivates older adults to volunteer, particularly as the Dutch population ages. However, in the discussion and conclusion, there is comparatively little time spent on the practical implications of the findings. The case for the research implications is pretty clear, but for a study that grounded in practice, it would be appropriate to at least touch on how these findings can actually influence the practice of recruiting and retaining Dutch volunteers aged 60-80.
The overall design of the study is strong, and fairly well-explained by the authors. I appreciate the overview of the Lifelines cohort study, and how this add-on contributes to its overall goals. The justification for selecting the VFI instrument, and the procedure for its translation, were done well. The implementation of the EFA and CFA procedures was appropriate. One very minor clarification: I found it curious that the authors chose to use SPSS for the EFA and STATA for the CFA.
I the text, you indicate that item 22 did not load onto any factors; however, in Table 3, you provide a loading for item 22, but not item 16. Is this an error in the table?
In lines 304-305, you indicate that the “findings support the Dutch translation of the VFI as a measure of volunteer motivations.” I recommend more specificity here to clearly indicate that the translation is a measure of volunteer motivations in the sample of volunteers aged 60+.
The article requires editing for grammar throughout.
Author Response
Points and Responses
Point 1: In the introduction to the article, the authors do a nice job of framing the need for the study by focusing on the practical importance of learning more about what motivates older adults to volunteer, particularly as the Dutch population ages. However, in the discussion and conclusion, there is comparatively little time spent on the practical implications of the findings. The case for the research implications is pretty clear, but for a study that grounded in practice, it would be appropriate to at least touch on how these findings can actually influence the practice of recruiting and retaining Dutch volunteers aged 60-80.
Response 1: We thank the reviewer for this suggestion. We agree that the validated scale can be useful for retention and recruitment practices and we now elaborate a bit on this in the Discussion part. Hereby we focus on the retention practices, as we have validated the VFI in a sample of current volunteers. Future research should show whether the VFI can be used to assess motives of potential volunteers, too.
We have added the following sentence to the Discussion part (see lines 411-418):
“The current study demonstrates that the Dutch version of the VFI (6 scales, 27 items) is a valid instrument for assessing volunteer motives among Dutch volunteers aged 60 years and over. Volunteer motives of Dutch older volunteers can now be assessed by making use of a validated measurement instrument, enabling researchers to assess and compare motives of Dutch older volunteers. The possibility to improve the knowledge and understanding of volunteer motives in Dutch older volunteers, could be helpful for volunteer retention practices. Voluntary work could be arranged in such a way that volunteer activities more closely address the motives driving Dutch older volunteers, potentially improving retention rates.”
Point 2: One very minor clarification: I found it curious that the authors chose to use SPSS for the EFA and STATA for the CFA.
Response 2: We thank the reviewer for this question. SPSS is used and supported at our department and is therefore used in general for all analyses. In SPSS however, no Confirmatory Factor Analyses can be performed. For that reason, we have used STATA for these specific analyses.
Point 3: In the text, you indicate that item 22 did not load onto any factors; however, in Table 3, you provide a loading for item 22, but not item 16. Is this an error in the table?
Response 3: We thank the reviewer for noticing this mistake, there was indeed an error in the table. The factor loading that was presented for item 22, belonged to the item presented above item 22 (item 19). We have corrected this mistake in the table in the manuscript(see page 15).
We have moreover presented the correct final table, without using track changes, at the end of this cover letter, to be sure that the numbers are presented at the correct position, because accepting the tracked changes in the manuscript could result in up- or downwards movement of the numbers again.
Point 4: In lines 304-305, you indicate that the “findings support the Dutch translation of the VFI as a measure of volunteer motivations.” I recommend more specificity here to clearly indicate that the translation is a measure of volunteer motivations in the sample of volunteers aged 60+.
Response 4: We thank the reviewer for this comment and have changed the sentence according to the reviewers’ suggestion (see Lines 306-307):
“Our findings offer support for the Dutch translation of the VFI as a measure of volunteer motivations among samples of volunteers aged 60 years and over.” (the newly added text is the part in Italics).
Point 5: The article requires editing for grammar throughout.
Response 5:
We thank the reviewer for this suggestion and we (two of the authors of this paper) have extensively checked and corrected the entire manuscript.
This manuscript is a resubmission of an earlier submission. The following is a list of the peer review reports and author responses from that submission.
Round 1
Reviewer 1 Report
ABSTRACT
The abstract is very well written. I would add a sentence about the need for a Dutch version of this scale.
INTRODUCTION
Need statistics on aging in the Netherlands. Also, we need information on the types of volunteering and the populations that are at the receiving end of volunteering. What has research show are the benefits of volunteering?
The Functional Approach – is that a theory? If so, please explain the main concepts and how they connect to the scale. A diagrammatic representation would be helpful here.
METHODS
Are explained in detail and comprehensively.
RESULTS
Are explained in excellent detail.
DISCUSSION
It would be better to first begin by summarizing which items were found to be valid and which items from the English version did not correspond in the Dutch version.
Then discuss why the items that could not be validated in the Dutch version.
LIMITATIONS
Was the sample tested for cultural differences, religious affiliation, immigrant status, etc.?
I believe we need a section on implications and moving forward.
Reviewer 2 Report
This study reported the psychometric properties of VFI Dutch version. The topic would be relevant and important to this journal. However, as indicated in major comments, the manuscript seems to have several concerns about the results.
<Major Comments>
1. Is it reasonable to include “Career” subscale in VFI Dutch version? The items of this subscale were remarkably skewed. As shown in Table 2, for the all items of this subscale, the indices of skewness and kurtosis were higher than 2.0 and mean scores were around 1.5. The factor mean score of Career was 1.49. Although the discussion section speculated potential reasons (lines 328 to 355), these extremely skewed data implies that the content validity of this subscale would not be sufficient among your populations. More detailed justification seems to be necessary.
2. Did the results of this study support the factorial validity of the new scale? The model fit indices of CFA and TLI did not reach the adequate levels (lines 181 to 182), especially with ADF estimation method. In the Discussion section, please compare the specific values of the model fit indices between this study and previous studies, and give comments on the factorial validity of the scale.
3. For the results of the sensitivity analyses, some indices were remarkably decreased from ML to ADF estimation method (i.e., CFI, TLI, SRMR). These results would not support the robustness of the model. Furthermore, it would be helpful for readers to give comments on the results that while RMSEA was decreased from ML to ADL estimation method, SRMR was increased.
4. As above comments, the validities of the new scale were not adequately supported. Thus, more conservative conclusions about the validities would be required for this study. Current conclusions (lines 377 to 380) might not be based on the results.
<Minor Comments>
5. Throughout the Method section, please grammatically check the term “will”. I am not a native speaker of English. However, I think that some parts using this terms would not be grammatically appropriate.
6. In the CFA section (lines 175 to 182), please indicates the cut-off points and brief explanations of SRMR and CD.
7.As the first use, please spell out ADF in line 188.
8. In the limitation section, the lack of test-retest reliability should be indicated as the additional limitation.